# WorldComp2D: Spatio-semantic Representations of Object Identity and Location from Local Views

**SeongMin Jin** [1]   **Doo Seok Jeong** [1 2]

## Abstract

Learning latent representations that capture both semantic and spatial information is central to efficient spatio-semantic reasoning. However, many existing approaches rely on implicit latent structures combined with dense feature maps or task-specific heads, limiting computational efficiency and flexibility. We propose World-Comp2D, a novel lightweight representation learning framework that explicitly structures latent space geometry according to object identity and spatial proximity using multiscale *local* receptive fields. This framework consists of (i) a proximity-dependent encoder that maps a given observation into a spatio-semantic latent space and (ii) a localizer that infers the coordinates of objects in the input from the resulting spatio-semantic representation. Using facial landmark localization as a proof-of-concept, we show that, compared to SoTA lightweight models, WorldComp2D reduces the numbers of parameters and FLOPs by up to $4.0\times$ and $2.2\times$, respectively, while maintaining real-time performance on CPU. These results demonstrate that explicitly structured latent spaces provide an efficient and general foundation for spatio-semantic reasoning. This framework is open-sourced at https://github.com/JinSeongmin/WorldComp2D.

## 1. Introduction

Embodied artificial intelligence (AI) agents may need to rely on internal latent representations to encode high-dimensional and complex real-world visual input into compact, structured forms that summarize the underlying fac-

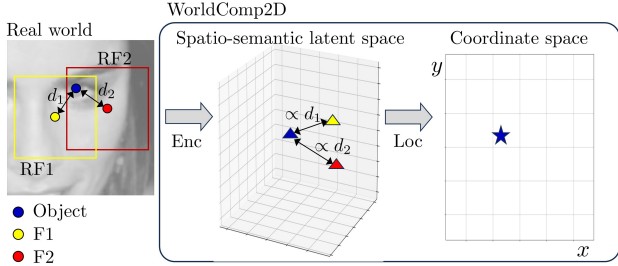

*Figure 1.* Overview of WorldComp2D. Observations made by an agent are encoded into a spatio-semantic latent space in which object identity is preserved and latent distances reflect real-world spatial proximity. Object locations are then inferred from these representations via the localizer. RF1 and RF2 denote two receptive fields centered at fixation points F1 and F2, respectively.

tors of variation (Ha & Schmidhuber, 2018). Unlike conventional vision systems, embodied agents operate under fundamentally different conditions: instead of having access to a complete global view of the environment, they perceive the world through local observations dependent on viewpoints (Papoudakis et al., 2021). To operate in real-time, such agents must rapidly construct internal representations that support instance-level understanding from these local views while operating under strict constraints on computation and power consumption.

Despite significant progress in representation learning, most existing approaches to spatio-semantic inference, which require jointly reasoning about semantic identity and spatial location such as object localization, are designed around global image access. These methods typically process full-images through pixel-wise computation (Ren et al., 2016; Lin et al., 2017), specialized architectures (Newell et al., 2016), or heatmap-based regression heads (Wang et al., 2020; Huang et al., 2021). Although effective in standard computer vision benchmarks, they fundamentally assume a complete global view of the environment, which conflicts with embodied AI settings. Moreover, full-image processing incurs substantial computational cost and latency, which limits their suitability for real-time deployment constraints.

In this work, we propose WorldComp2D, a novel lightweight representation learning framework for embod-

[1]Division of Materials Science and Engineering [2]Department of Semiconductor Engineering, Hanyang University, Republic of Korea. Correspondence to: Doo Seok Jeong <dooseokj@hanyang.ac.kr>.

ied AI. Our key idea is to directly encode spatio-semantic structure into the latent space. As illustrated in Fig. 1, observations made by an agent are encoded into a latent space where object identity is preserved and latent distances reflect real-world spatial proximity. Then, a localizer predicts object locations directly from these representations. This design enables efficient spatio-semantic inference from a small set of local observations, consistent with the perceptual and computational constraints of embodied agents.

We evaluate WorldComp2D using facial landmark localization as a proof-of-concept. While achieving slightly lower accuracy than SoTA regression methods, our framework enables robust inference from limited visual observations with substantially reduced computational complexity. WorldComp2D requires only $2.4$M parameters and $< 550$ MFLOPs, achieving $> 78$ frames per second (FPS) on CPU, whereas prior approaches typically rely on significantly larger models with $9.7 - 67$M parameters and $1.2 - 26.8$ GFLOPs.

Our contributions are as follows:

- We propose WorldComp2D, a lightweight fixation-centered framework that learns spatio-semantic representations from local views and localizes objects using an encoder and a localizer, with an optional refinement module.

- We design a learning objective that preserves object identity while encouraging latent-space distances to reflect real-world distances to the fixation point through proximity-weighted contrastive learning.

- We demonstrate that WorldComp2D achieves competitive localization accuracy with real-time efficiency and flexible accuracy–speed trade-offs, supported by extensive ablation and robustness studies.

## 2. Related Work

### 2.1. Representation Learning for Embodied AI

In embodied AI, latent representations serve as internal states that summarize sensory observations under partial observability, as agents perceive the environment through local, viewpoint-dependent inputs. In this context, previous work has explored world model-based approaches that map high-dimensional observations to compact latent states, where a learned transition function predicts their evolution over time for imagination, planning, and control (Ha & Schmidhuber, 2018; Hafner et al., 2019). These approaches enable agents to predict how the environment changes over time under their actions. Although effective for sequential decision making, the latent space is primarily shaped by temporal prediction objectives and is not explicitly structured to

preserve fine-grained spatial or instance-level information.

In contrast, WorldComp2D is designed to directly support embodied perception by explicitly structuring latent space geometry. Rather than focusing on temporal dynamics, our framework encodes the semantic identity of objects and fixation-centered spatial proximity as intrinsic properties of the representation. By embedding local view-dependent semantic and spatial information directly into latent space, WorldComp2D enables efficient spatio-semantic inference from a small set of local observations, providing an internal representation that is naturally aligned with embodied perception.

### 2.2. Contrastive Learning

Contrastive learning has emerged as a powerful paradigm for learning discriminative representations by encouraging similarity between related samples while pushing apart unrelated ones. In a typical setting, two augmented views are generated for each sample, resulting in $2N$ views for a mini-batch of $N$ samples. Based on these augmented views, contrastive learning methods define binary positive and negative pairs according to instance identity, data augmentations, or semantic labels (Chen et al., 2020; Tian et al., 2020; Khosla et al., 2020).

Our approach departs from conventional binary contrastive formulations by modeling pairwise relationships with continuous affinity values. Rather than assigning equal importance to all positive pairs, we incorporate spatial proximity as a continuous signal that modulates the strength of pairwise interactions. This design enables a more precise and fine-grained notion of distance in the latent space, allowing representations to capture fixation-centered proximity beyond categorical similarity.

### 2.3. Object Localization

Object localization has been widely studied for identifying and localizing objects or keypoints in images. Early approaches depend on full-image processing and pixel-wise prediction schemes, such as anchor-based detectors (Ren et al., 2016; Lin et al., 2017). While effective, these approaches often involve high computational cost and strong task-specific architectural assumptions.

Facial landmark localization can be viewed as a structured instance of object localization, in which semantically meaningful keypoints are estimated under fixed spatial constraints (Zhang et al., 2014; Bulat & Tzimiropoulos, 2017). Most existing facial landmark localization methods employ heatmap-based regression as the primary mechanism for spatial inference (Wang et al., 2019; 2020), typically requiring high-resolution feature maps and multi-stage refinement, which further increases computational cost.

WorldComp2D enables object localization without full-image processing. Rather than performing pixel-wise computations, our framework encodes a small set of local observations into latent representations and infers object locations by aggregating the resulting information directly in latent space. This design allows localization to be achieved efficiently from limited local views, without relying on exhaustive image-level processing. Conventional heatmap-based regression is optionally employed as a lightweight refinement mechanism, providing improved accuracy.

## 3. Methodology

WorldComp2D comprises a proximity-dependent encoder (PdEnc) and a localizer (Loc), which consider objects belonging to a set of given classes. WorldComp2D also includes an optional auxiliary localizer (AuxLoc) to improve object localization accuracy. These three models are illustrated in Fig. 2.

### 3.1. Proximity-dependent Encoder

**Local observation as input.** An AI agent is assumed to observe the 2D world through two receptive fields, $\mathrm{RF}^{[1]}$ and $\mathrm{RF}^{[2]}$, of different sizes but centered at the same fixation point $\boldsymbol{F} \in \mathbb{R}^2$, as illustrated in Fig. 3. This design ensures that each observation captures both fine-grained appearance cues and broader contextual information. Specifically, the observation $\boldsymbol{o}$ is constructed by extracting two patches from an input image of size $C \times H \times W$:

$$o^{[1]} \in \mathbb{R}^{C \times a \times a} \quad \text{from} \quad \mathrm{RF}^{[1]},$$
$$o^{[2]} \in \mathbb{R}^{C \times sa \times sa} \quad \text{from} \quad \mathrm{RF}^{[2]},$$

where $s > 1$ and $a \ll H, W$. The larger-scale patch $o^{[2]}$ is resized to $C \times a \times a$ and concatenated with $o^{[1]}$ along the channel dimension to form the final observation $\boldsymbol{o} \in \mathbb{R}^{2C \times a \times a}$. In this work, we set $a = 27$ and $s = 4$. When a receptive field extends beyond the image boundary, constant-value padding is applied to the out-of-bound regions.

**Spatio-semantic latent vector as output.** PdEnc encodes observation $\boldsymbol{o} \in \mathbb{R}^{2C \times a \times a}$ for a given fixation point $F$ into a spatio-semantic latent vector $\boldsymbol{z} \in \mathbb{R}^{d_z}$, where $d_z = 256$. This spatio-semantic latent vector lies on a hypersphere, $\boldsymbol{z}^\mathrm{T} \boldsymbol{z} = 1$.

**PdEnc architecture.** The network architecture of PdEnc is illustrated in Fig. 2**a**. PdEnc is a simple convolutional neural network (CNN) with six convolutional layers and two fully connected layers: 2C32(6C32)-32C32-32C64-64C64-64C128-128C256-FC512-FC256-L2Norm for grayscale(RGB) images. The final L2Norm operation constrains latent representations to lie on a hypersphere, which stabilizes learning and facilitates distance-based reasoning in the spatio-semantic latent space.

**Multiscale receptive fields.** The spatial proximity is defined as the real-world distance between the fixation point and surrounding objects within the second-scale receptive field $\mathrm{RF}^{[2]}$. This distance provides a continuous measure of spatial closeness, enabling PdEnc to learn latent representations that reflect both semantic similarity and fixation-centered spatial structure. It is important to note that spatial proximity can only be reliably inferred within the support of the receptive field since any distal objects outside the receptive field do not provide sufficient evidence for estimating spatial proximity.

**Proximity-weighted contrastive learning.** To train PdEnc to explicitly enforce the semantic and spatial structures described above, we propose a proximity-weighted contrastive learning method. This method uses multiple image samples, each of which includes multiple objects $c$ with their coordinates $\boldsymbol{x}_c$ annotated.

$$\mathbf{C}_i = \{c | \text{objects } c \text{ included in image } i\}.$$

For training, each mini-batch $\mathcal{B}$ is constructed by (i) randomly sampling images from a given dataset, (ii) randomly sampling a single object $c$ (along with their annotated coordinates $\boldsymbol{x}_c$) in $\mathbf{C}_i$ for each image, and (iii) subsequently constructing observation $\boldsymbol{o} \in \mathbb{R}^{2C \times a \times a}$ by placing the fixation point $F$ on the object coordinate $\boldsymbol{x}_c$ ($F = \boldsymbol{x}_c$) for each sampled object $c$. For the $i$th sample (object $c_i$ and $F_i = \boldsymbol{x}_{c(i)}$), we define a set of proximal objects:

$$\mathbf{P}_i = \left\{ c \in \mathbf{C}_i | \boldsymbol{x}_c \in o^{[2]} \text{ for } F_i \text{ on image } i \right\}. \quad (1)$$

An augmented mini-batch $\mathcal{B}'$ is constructed by (i) randomly placing a single fixation point $F$ on each image in the batch $\mathcal{B}$ and (ii) subsequently constructing observation $\boldsymbol{o}$ for each image. A set of proximal objects for each sample in $\mathcal{B}'$ is also defined as in Eq. (1).

The goal of proximity-weighted contrastive learning is two-fold:

- Optimal mapping of objects with regard to their identities and spatial proximity using the observations (with $F = \boldsymbol{x}_c$) in batch $\mathcal{B}$

- Optimal mapping of random observations with respect to their proximal objects using the observations (with random $F$) in augmented batch $\mathcal{B}'$

To this end, we introduce proximity-weighted contrastive

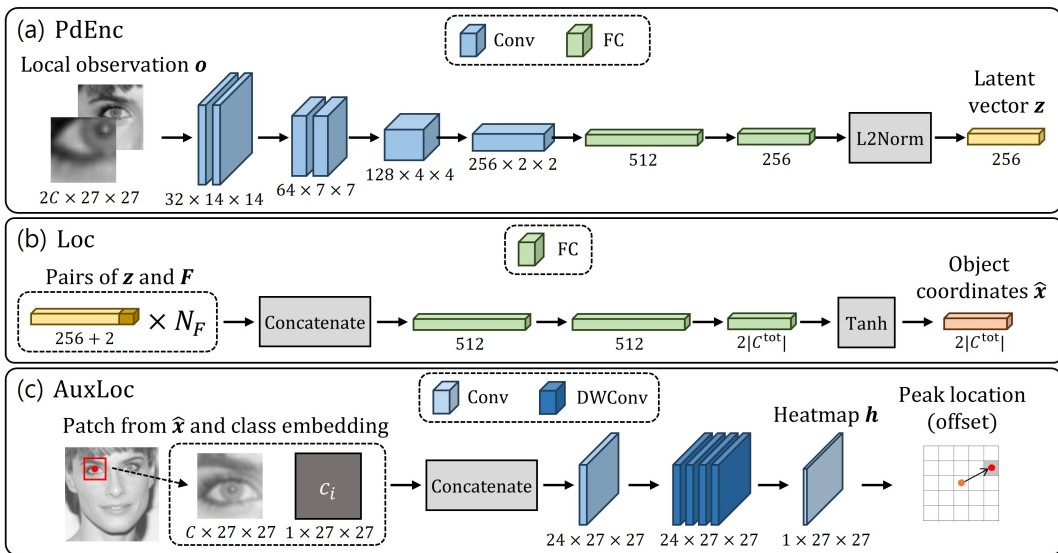

*Figure 2.* Networks in WorldComp2D. (**a**) Proximity-dependent encoder (PdEnc), which maps fixation-centered observations to a normalized latent vector. (**b**) Localizer (Loc), which aggregates paired latent vectors and fixation coordinates to predict object locations. (**c**) Auxiliary localizer (AuxLoc), an optional refinement module that estimates a heatmap from a local patch and class-conditioned embedding.

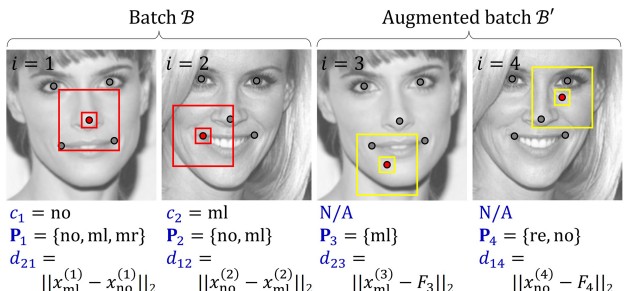

*Figure 3.* Example of sample augmentation for proximity-weighted contrastive learning. The right eye, left eye, nose, left mouth corner, and right mouth corner are denoted by `re`, `le`, `no`, `ml`, and `mr`, respectively.

loss (PWConLoss) as follows.

$$
\mathcal{L}_{\mathrm{PWC}} = \frac{-1}{|\mathcal{B}|} \sum_{i \in \mathcal{B}} \Bigg[ \underbrace{\frac{1}{N_i} \sum_{j \in \mathcal{B} \setminus \{i\}} w_{ij} \mathbb{1}_{\{c_i \in \mathbf{P}_j\}} l_{ij}}_{\text{between } c_i \text{ and } c_j \ (i \neq j)}
$$
$$
+ \underbrace{\frac{1}{N_i'} \sum_{j \in \mathcal{B}'} w_{ij} \mathbb{1}_{\{c_i \in \mathbf{P}_j\}} l_{ij}}_{\text{between } c_i \text{ and random observation } \boldsymbol{o}} \Bigg]. \tag{2}
$$

$$
N_i = \sum_{j \in \mathcal{B} \setminus \{i\}} \mathbb{1}_{\{c_i \in \mathrm{P}_j\}}, \quad N_i' = \sum_{j \in \mathcal{B}'} \mathbb{1}_{\{c_i \in \mathrm{P}_j\}},
$$

$$
l_{ij} = \log \frac{\exp(\boldsymbol{z}_i^{\mathrm{T}} \boldsymbol{z}_j / \tau)}{\sum_{k \in (\mathcal{B} \cup \mathcal{B}') \setminus \{i\}} \exp(\boldsymbol{z}_i^{\mathrm{T}} \boldsymbol{z}_k / \tau)},
$$

where $\boldsymbol{z}_i$ and $\tau$ denote the output vector for the $i$th sample and temperature parameter, respectively. The weight $w_{ij}$ is constrained to range $(1, 2]$ such that

$$
w_{ij} = 1 + \exp(-0.025 d_{ij}), \tag{3}
$$

where $d_{ij} = ||\boldsymbol{x}_{c(i)}^{(j)} - F_j||_2$. Note that $\boldsymbol{x}_{c(i)}^{(j)}$ means the coordinate of object $c_i$ on the $j$th sample. The constant 0.025 is chosen so that $w_{ij} \approx 1.5$ when $d_{ij} = 27$, which corresponds to the height and width of the smaller receptive field $o^{[1]}$ with size $C \times 27 \times 27$. Examples of sample augmentation for proximity-weighted contrastive learning are shown in Fig. 3.

### 3.2. Localizer

**Localizer input.** Localizer (Loc) infers the coordinates of proximal objects in P in Eq. (1) from the spatio-semantic output $\boldsymbol{z}$ of PdEnc for a given fixation point $F$. Assume that all objects in a given dataset ($\mathrm{C}^{\mathrm{tot}} = \bigcup_{i=1}^{N_s} \mathrm{C}_i$; $N_s$ is the total number of samples) are included in a set of proximal objects for any fixation point. If this holds, a single observation is sufficient to localize all objects. However, since observation is made through *local* receptive fields over an entire image, this assumption unlikely holds. Therefore, Loc collects $N_o$ observations (for different fixation points) that, in aggregate, cover all object classes in $\mathrm{C}^{\mathrm{tot}}$. A pair of a fixation-point and the resulting latent representation $(\boldsymbol{z}, F)$ is concatenated across total $N_o$ observations and subsequently vectorized to construct Loc input $\boldsymbol{I} \in \mathbb{R}^{N_o(d_z+2)}$.

**Localizer architecture.** The network architecture of Loc is

illustrated in Fig. 2**b**. Loc is designed to simultaneously predict the object coordinates normalized to the range $(-1, 1)$ for all $|\mathrm{C}^{\mathrm{tot}}|$ object classes, $\hat{\boldsymbol{x}} \in \mathbb{R}^{2|\mathrm{C}^{\mathrm{tot}}|}$. It is a simple multilayer perceptron of FC512-FC512-FC$n$-Tanh, where $n = 2|\mathrm{C}^{\mathrm{tot}}|$. Rather than producing independent predictions per observation, Loc jointly aggregates all latent representations to infer object coordinates for each class. By leveraging the spatio-semantic structure encoded in $\boldsymbol{z}$, Loc enables localization of class-specific objects from a small set of local observations, without relying on pixel-wise computation or global image access.

### 3.3. Auxiliary Localizer

**Auxiliary localizer input.** Auxiliary localizer (AuxLoc) is optional model that refines the object coordinates $\hat{\boldsymbol{x}}$ inferred by Loc for each object separately. For the predicted coordinate $\hat{\boldsymbol{x}}$ in each object class, a first-scale image patch $o^{[1]} \in \mathbb{R}^{C \times a \times a}$ for $F = \hat{\boldsymbol{x}}$ is extracted from the input image. An object class-conditioned embedding is constructed as a tensor of size $1 \times a \times a$, with values normalized to the range $[-1, 1]$, and concatenated with $o^{[1]}$ along the channel dimension, yielding AuxLoc input $\boldsymbol{I} \in \mathbb{R}^{(C+1) \times a \times a}$.

**Auxiliary localizer architecture.** The network architecture of AuxLoc is illustrated in Fig. 2**c**. AuxLoc is a CNN of 2C24(4C24)-4×DW24-C1 for grayscale(RGB) images, where DW denotes a depth-wise separable convolution block (Chollet, 2017). In response to the local view centered at $\hat{\boldsymbol{x}}$, AuxLoc outputs a class-specific heatmap $\boldsymbol{h} \in \mathbb{R}^{1 \times a \times a}$. The refined object location is obtained by applying an offset derived from the peak of the heatmap $\boldsymbol{h}$ to the corresponding coarse estimate $\hat{\boldsymbol{x}}$.

## 4. Experimental Results

As a proof-of-concept, we applied the WorldComp2D framework to facial landmark localization tasks in which facial landmarks are the objects whose spatio-semantic representations are learned. We used the COFW (Burgos-Artizzu et al., 2013), 300W (Sagonas et al., 2016), and AFLW (Koestinger et al., 2011) datasets. The COFW dataset consists of 1,345 training and 507 test grayscale images, each annotated with 29 facial landmarks ($|\mathrm{C}^{\mathrm{tot}}| = 29$). COFW frequently contains facial landmarks occluded by external objects. The 300W dataset contains 3,148 training and 689 test RGB images, each annotated with 68 landmarks ($|\mathrm{C}^{\mathrm{tot}}| = 68$), and exhibits substantial variations in facial pose and illumination conditions. The AFLW dataset comprises 20,000 training and 4,386 test RGB images, each annotated with 19 landmarks ($|\mathrm{C}^{\mathrm{tot}}| = 19$), and is characterized by large pose variations and partial facial visibility.

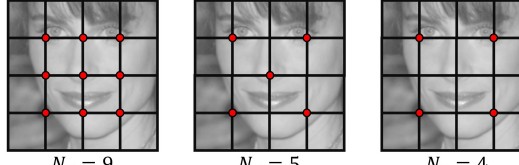

*Figure 4.* Fixation points on a given image for $N_{\mathrm{F}} = 9$, 5, and 4.

### 4.1. Implementation Detail

Each image was cropped to include the full head, randomly rescaled ($\pm 5\%$), horizontally flipped ($50\%$), and rotated ($60\%, \pm 10°$), then resized to $256 \times 256$. During facial landmark localization, we first computed the mean location for each landmark across the samples in a given dataset, and Loc predicted an offset relative to this mean. To improve robustness, random offsets uniformly sampled from $[-5, 5]$ pixels were applied to the fixation points when training Loc and AuxLoc. The ground-truth heatmap for AuxLoc is constructed as a 2D Gaussian centered at the ground-truth landmark coordinate within the extracted patch $o^{[1]}$, with a fixed standard deviation of 1.5. The samples whose ground-truth coordinates fall outside the extracted patch $o^{[1]}$ were ignored during training.

We set the number of fixation points to nine ($N_{\mathrm{F}} = 9$) to cover all landmarks on each $256 \times 256$ sample, evenly distributed at fixed spatial intervals of 64 pixels along both image axes. Examples of $N_{\mathrm{F}} = 9/5/4$ are shown in Fig. 4. The coordinate refinement by AuxLoc was constrained to at most 2 pixels on COFW and 1 pixel on other datasets. The models were trained using the Pytorch framework (Paszke et al., 2019) on a GPU workstation (RTX A6000; Xeon Gold CPU 2.9GHz; 256GB DRAM). The hyperparameters and learning behaviors of PdEnc, Loc, and AuxLoc are detailed in Appendix.

### 4.2. Analysis of proximity-dependent encoder

**Intra-class clustering.** We evaluated the L2 distances between individual spatio-semantic representations for each class on COFW (with 29 landmarks). Their mean values are plotted in Fig. 5**a**, indicating successful class-wise clustering. This result suggests that the latent space effectively preserves landmark identity, yielding stable landmark-specific representations across different spatial contexts.

**Inter-class separation.** Fig. 5**b** shows the L2 distances between the left pupil (Class 17) representation and other landmark representations within the same image. While clear separation is observed across landmark clusters, landmarks that are spatially adjacent to the left pupil, such as other left eye related landmarks, exhibit relatively smaller distances. This pattern indicates that the latent space distin-

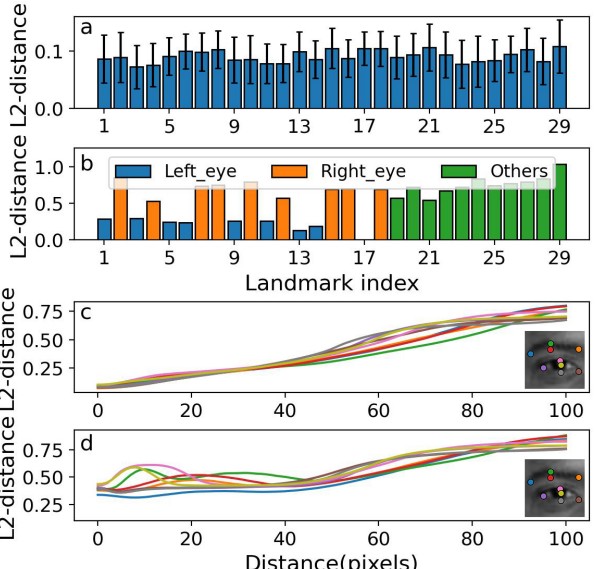

*Figure 5.* Analysis of spatio-semantic latent representation. (**a**) L2 distances between individual landmark representations and their corresponding class means. (**b**) L2 distances between the left pupil representation and other landmarks representations within the same image. L2 distances as a function of spatial distance from a given landmark obtained from (**c**) PdEnc and (**d**) from an encoder trained using proximity-*unweighted* contrastive loss.

guishes landmark identity while simultaneously encoding relative spatial relationships among landmarks.

**Distance-preserving embedding.** We examined whether the real-world distance between an object and the fixation point is preserved in the spatio-semantic latent space (i.e., whether latent-space distance reflects physical proximity) by sampling observations on a fixation point with controlled offsets relative to the left eye landmark. As shown in Fig. 5**c**, L2 distances increase with real-world distance, indicating that spatial proximity is preserved in the latent space. Within the range of the second-scale receptive field (approximately 54 pixels), this pattern remains consistent across landmarks. Beyond this range, real-world proximity can no longer be reliably inferred, resulting in increased landmark-dependent variability in latent distances.

As a counter part, we trained an encoder of the same architecture as PdEnc but using proximity-*unweighted* contrastive loss that is the same as Eq. (2) but with $w_{ij} = 1$ for all $i$ and $j$. Fig. 5**d** shows the latent space L2 distance between a given object and a fixation point with real-world distance. Compared with Fig. 5**c**, the spatial proximity is obviously unpreserved.

### 4.3. Results on Facial Landmark Localization

**Localization accuracy and efficiency.** Fig. 6 presents landmark localization results on several samples from COFW,

300W, and AFLW. The localization accuracy is measured using the Normalized Mean Error (NME), where the error is normalized by the inter-ocular distance ($NME_{IO}$) or inter-pupil distance ($NME_{IP}$) or the diagonal length of the face bounding box ($NME_{Diag}$), following standard evaluation protocols for each dataset. Since WorldComp2D is not trained via direct supervised regression for landmark localization, it yields higher NME values on COFW and 300W compared to regression-based SoTA methods listed in Table 1. In contrast, on the AFLW, our framework achieves lower NME than LAB (Wu et al., 2018), Awing (Wang et al., 2019), ODN (Zhu et al., 2019), and HRNet (Wang et al., 2020).

Beyond localization accuracy, WorldComp2D highlights extremely low computational complexity. The entire framework consists of 2.4M parameters in total, with 1.1M, 1.3M, and 4.0K parameters assigned to PdEnc, Loc, and AuxLoc, respectively. In terms of computational cost, PdEnc, Loc, and AuxLoc require approximately 15.7M, 3.0M, and 5.9M FLOPs, respectively. Since the framework processes nine fixation-centered observations ($N_F = 9$), the total number of FLOPs is computed as

$$FLOPs_{tot} = 9FLOPs_{PdEnc} + FLOPs_{Loc} + |C^{tot}|FLOPs_{AuxLoc},$$

resulting in 293.7, 546.8, and 256.9 MFLOPs on COFW, 300W, and AFLW, respectively. Compared to the lightweight PoPos (Xiang et al., 2025) model with 9.7M parameters, WorldComp2D achieves a $4.0\times$ reduction in the number of parameters and a $2.2\times$ reduction in FLOPs in the worst case.

**Real-time operation.** Due to its low computational complexity, our method runs at 138.41 (COFW), 78.48 (300W), and 163.55 (AFLW) frames per second (FPS) on a desktop (i5-13400 CPU 2.5GHz; 128GB DRAM), using a batch size of 1 (Table 2). This highlights the feasible real-time operation of WorldComp2D.

We decompose the overall framework into five sub-workloads: (1) patch extraction (local observation for a given fixation point for PdEnc), (2) PdEnc operation, (3) Loc operation, (4) optional patch extraction for AuxLoc, and (5) AuxLoc operation. Fig. 7 shows the normalized runtime for each of the five workloads. Across all datasets, AuxLoc occupies the largest contribution to the total runtime, followed by PdEnc and patch extraction stages. This is because AuxLoc (despite lightweight) processes a separate local patch for each landmark. Accordingly, its contribution decreases as the number of landmarks decreases. PdEnc accounts for the larger contribution than Loc as it operates on multiple fixation-centered patches per frame, whereas Loc relies on a single forward pass on aggregated representations. Patch extraction constitutes another non-negligible component since it involves patching and resizing for the

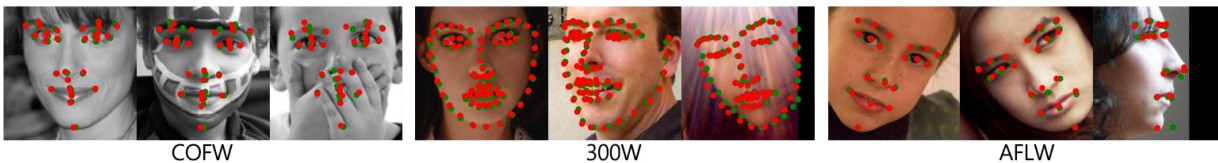

|  | COFW | 300W | AFLW |
|---|---|---|---|

*Figure 6.* Localized landmarks (red circles) and ground-truth annotations (green circles) on sample images from COFW, 300W and AFLW.

*Table 1.* Comparison of our method with SoTA approaches on COFW, 300W and AFLW.

| Method | COFW | | 300W | AFLW | # Params (M) | FLOPs |
|---|---|---|---|---|---|---|
| | NME$_{IO}$ | NME$_{IP}$ | NME$_{IO}$ | NME$_{Diag}$ | | |
| LAB (Wu et al., 2018) | 3.92 | 5.58 | 3.49 | 1.85 | 25.1 | 18.9G |
| AWing (Wang et al., 2019) | - | 4.94 | 3.07 | 1.53 | 24.2 | 26.8G |
| AVS (Qian et al., 2019) | - | 4.43 | 3.86 | 1.86 | 28.3 | 2.4G |
| ODN (Zhu et al., 2019) | - | - | 4.17 | 1.63 | - | - |
| HRNet (Wang et al., 2020) | 3.45 | - | 3.32 | 1.57 | 9.7 | 4.8G |
| PIP (Jin et al., 2021) | 3.45 | - | 3.19 | 1.42 | 45.7 | 10.5G |
| ADNet (Huang et al., 2021) | - | 4.69 | 2.93 | - | 13.4 | 17.0G |
| SDFL (Lin et al., 2021) | 3.63 | - | - | - | - | 5.2G |
| HIH (Lan et al., 2021) | 3.21 | 4.63 | 3.09 | - | 22.7 | 17.2G |
| SLPT (Xia et al., 2022) | 3.32 | 4.63 | 3.17 | - | 13.2 | 6.1G |
| STARLoss (Zhou et al., 2023) | - | 4.62 | 2.87 | - | 13.4 | - |
| RHT-R (Wan et al., 2023) | - | 4.42 | 2.82 | 1.18 | - | - |
| D-ViT (Dang et al., 2025) | - | 4.13 | 2.85 | - | 67.3 | 21.8G |
| PoPos (Xiang et al., 2025) | - | 3.80 | 3.28 | 1.43 | 9.7 | 1.2G |
| **WorldComp2D on COFW/300W/AFLW** | 5.16±0.05 | 7.43±0.07 | 5.06±0.01 | 1.52±0.01 | 2.4 | 293.7M / 546.8M / 256.9M |

*Table 2.* FPS and tolerance to reduced data precision.

| Precision | COFW | | 300W | | AFLW | |
|---|---|---|---|---|---|---|
| | FPS | NME$_{IO}$ | FPS | NME$_{IO}$ | FPS | NME$_{Diag}$ |
| FP32 on CPU i5 | 138.4 | 5.16±0.05 | 78.48 | 5.06±0.01 | 163.55 | 1.52±0.01 |
| FP32 on GPU A6000 | 9328.96 | 5.16±0.05 | 5905.59 | 5.06±0.01 | 6381.85 | 1.52±0.01 |
| FP16 on GPU A6000 | 9556.62 | 5.16±0.05 | 7217.09 | 5.06±0.02 | 7709.42 | 1.52±0.01 |
| BF16 on GPU A6000 | 9834.96 | 5.17±0.04 | 7486.92 | 5.07±0.02 | 7743.49 | 1.52±0.01 |

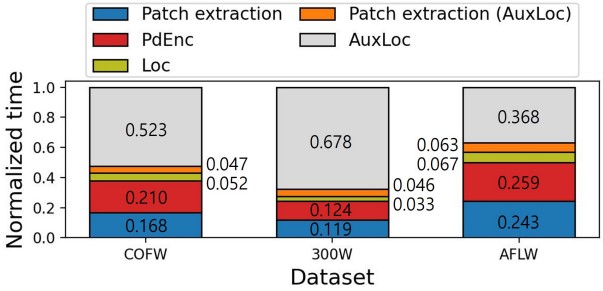

*Figure 7.* Normalized localization runtime decomposed into five sub-workloads.

*Table 3.* Effect of AuxLoc on localization accuracy and computational efficiency on CPU i5.

| | COFW | | |
|---|---|---|---|
| Model | NME$_{IO(Diag)}$ | FLOPs | FPS |
| with AuxLoc | 5.16±0.05 | 293.7M | 138.41 |
| w/o AuxLoc | 5.41±0.04 | 140.5M | 344.19 |
| | 300W | | |
| with AuxLoc | 5.06±0.01 | 546.8M | 78.48 |
| w/o AuxLoc | 5.22±0.01 | 144.7M | 282.34 |
| | AFLW | | |
| with AuxLoc | 1.52±0.01 | 256.9M | 163.55 |
| w/o AuxLoc | 1.61±0.01 | 144.6M | 304.93 |

second-scale receptive field ($o^{[2]}$). This leads to a longer processing time than patch extraction for AuxLoc that uses first-scale patches only.

As such, AuxLoc serves as an auxiliary module that refines the localization predicted by Loc, but it still accounts for

the largest contribution to the overall runtime. Table 3 reports the accuracy improvement from adding AuxLoc and its computational overhead. In the best case, AuxLoc yields only a 4.6% reduction in NME on COFW, underscoring the strong localization accuracy of the baseline (Loc-only) model. However, omitting AuxLoc during object local-

*Table 4.* Localization accuracy under various degradations.

| Degradation | COFW | 300W | AFLW |
|---|---|---|---|
| Baseline | 5.16±0.05 | 5.06±0.01 | 1.52±0.01 |
| Blur ($\sigma = 1$) | 5.15±0.05 | 5.06±0.02 | 1.53±0.01 |
| Blur ($\sigma = 2$) | 5.27±0.04 | 5.17±0.04 | 1.57±0.01 |
| Blur ($\sigma = 3$) | 5.60±0.02 | 5.43±0.06 | 1.64±0.02 |
| JPEG ($Q = 80$) | 5.17±0.04 | 5.08±0.01 | 1.52±0.01 |
| JPEG ($Q = 60$) | 5.18±0.03 | 5.10±0.01 | 1.52±0.01 |
| JPEG ($Q = 40$) | 5.19±0.03 | 5.11±0.01 | 1.53±0.01 |
| JPEG ($Q = 20$) | 5.22±0.06 | 5.19±0.02 | 1.54±0.01 |
| Motion Blur ($k = 5$) | 5.17±0.06 | 5.09±0.03 | 1.53±0.01 |
| Motion Blur ($k = 10$) | 5.57±0.03 | 5.42±0.04 | 1.68±0.01 |
| Occlusion (size $= 20$) | 5.33±0.05 | 5.27±0.01 | 1.59±0.01 |
| Occlusion (size $= 40$) | 5.56±0.07 | 5.58±0.01 | 1.64±0.01 |

*Table 5.* PdEnc with different model complexities.

| COFW | | | |
|---|---|---|---|
| Model | # Params | $\text{NME}_{\text{IO(Diag)}}$ | FLOPs |
| PdEnc (C9) | 1.4M | 5.80±0.05 | 162.3M |
| PdEnc (C16) | 1.8M | 5.30±0.06 | 195.2M |
| PdEnc (C32) | 2.4M | 5.16±0.05 | 293.7M |
| 300W | | | |
| PdEnc (C9) | 1.4M | 5.88±0.07 | 412.5M |
| PdEnc (C16) | 1.8M | 5.28±0.03 | 446.2M |
| PdEnc (C32) | 2.4M | 5.06±0.01 | 546.8M |
| AFLW | | | |
| PdEnc (C9) | 1.4M | 1.74±0.02 | 122.6M |
| PdEnc (C16) | 1.8M | 1.57±0.01 | 156.4M |
| PdEnc (C32) | 2.4M | 1.52±0.01 | 256.9M |

*Table 6.* Cross-dataset evaluation result.

| Model | 300W | COFW-68 | Degradation |
|---|---|---|---|
| LAB (Wu et al., 2018) | 3.49 | 4.62 | 32.4% |
| ODN (Zhu et al., 2019) | 4.17 | 5.30 | 27.1% |
| PIP (Jin et al., 2021) | 3.23 | 4.23 | 31.0% |
| WorldComp2D | 5.06±0.01 | 6.08±0.14 | 20.2% |

*Table 7.* Localization accuracy and efficiency with different numbers of fixation points.

| COFW | | | |
|---|---|---|---|
| Model | # Params | $\text{NME}_{\text{IO(Diag)}}$ | FLOPs |
| $N_{\text{F}} = 4$ | 1.6M | 5.75±0.06 | 216.0M |
| $N_{\text{F}} = 5$ | 1.8M | 5.30±0.06 | 231.6M |
| $N_{\text{F}} = 9$ | 2.4M | 5.16±0.05 | 293.7M |
| 300W | | | |
| $N_{\text{F}} = 4$ | 1.7M | 5.60±0.04 | 466.8M |
| $N_{\text{F}} = 5$ | 1.9M | 5.34±0.04 | 482.8M |
| $N_{\text{F}} = 9$ | 2.4M | 5.06±0.01 | 546.8M |
| AFLW | | | |
| $N_{\text{F}} = 4$ | 1.7M | 1.69±0.01 | 176.9M |
| $N_{\text{F}} = 5$ | 1.9M | 1.62±0.01 | 192.9M |
| $N_{\text{F}} = 9$ | 2.4M | 1.52±0.01 | 256.9M |

ization substantially reduces FLOPs and correspondingly improves FPS. This FPS gain becomes more pronounced as the number of landmarks increases, since AuxLoc causes additional per-landmark computation.

**Effect of data precision.** We examined the localization accuracy and computational efficiency for three different data formats (FP32, FP16, and BF16) on GPU. As summarized in Table 2, FP16 and BF16 substantially improve FPS while leading to only negligible degradation in NME. These results highlight that WorldComp2D can be deployed efficiently under practical deployment constraints.

**Robustness to input degradation.** Since PdEnc encodes real-world objects, robustness to input degradations is a prerequisite for reliable downstream localization. To identify this robustness, we evaluated the localization performance of WorldComp2D under various degradation cases, including Gaussian blur, JPEG degradation, motion blur, and occlusion. For the occlusion experiments, a square patch (height=width=size) with zero-valued pixels was inserted at a random location on each test sample. Table 4 summarizes the results. Across all datasets and degradation types, WorldComp2D exhibits consistently moderate performance degradation. Notably, the largest absolute degradation is observed under extreme occlusion (size=40) on 300W, where NME increases by 10.2% only. These results suggest that our spatio-semantic representations provide a stable foundation for robust inference under diverse real-world image degradations.

### 4.4. Ablation study

**PdEnc complexity versus accuracy.** We investigated a relationship between PdEnc complexity (model size) and the downstream localization accuracy. To this end, we considered two additional lighter models (C9 and C16) to our PdEnc (referred to as C32). The architectural details for these models are addressed in Appendix. The localization accuracy and the number of total FLOPs with these models are compared with C32 in Table 5. The downstream localiza-

tion accuracy tends to increase with the model complexity at the cost of a increase in computation number. Regarding this tradeoff, we chose C32 as our standard PdEnc that realizes competitive localization accuracy while remaining lightweight.

**Cross-dataset evaluation.** The COFW, 300W, and AFLW datasets contain different numbers of annotated landmarks. For cross-dataset evaluation, we conducted zero-shot landmark localization on COFW-68 (Ghiasi & Fowlkes, 2014) (COFW with 68 landmarks comparable to 300W with $|\text{C}^{\text{tot}}| = 68$) using the WorldComp2D framework trained on 300W. Table 6 reports the cross-validation result and several SoTA results. These regression-based SoTA methods suffer from a performance degradation of 27–32%, whereas WorldComp2D shows only a 20.2% increase in NME.

**Number of fixation points.** Loc uses the spatio-semantic representations for multiple fixation points ($N_{\text{F}} = 9$) in

aggregate as its input because a single *local* observation cannot include all objects (landmarks) in a given dataset. As such, we applied $3 \times 3$ periodic fixation points on each input image in an attempt to cover all landmarks for each set of nine observations. We also evaluated Loc with $N_F = 4$ and 5 as illustrated in Fig. 4. Table 7 summarizes the effect of $N_F$ on the localization accuracy on the three datasets and the number of total FLOPs. An increase in $N_F$ increases evidence for real-world objects to be mapped into the latent space, leading to a higher localization accuracy. Nevertheless, WorldComp2D achieves competitive localization accuracy even with $N_F = 4$, while significantly reducing its computational complexity.

## 5. Conclusion

We propose WorldComp2D, a lightweight representation learning framework for embodied AI agents that must reason from local observations under limited compute. It encodes local views into a spatio-semantic latent space that jointly captures object identity and spatial proximity, enabling localization without exhaustive full-image processing and providing an efficient alternative to global-view architectures.

Using facial landmark localization as a proof-of-concept, we show that this latent space has meaningful geometry (intra-class clustering, inter-class separation, and distances consistent with real-world proximity). WorldComp2D achieves competitive accuracy and high inference speed with substantially reduced compute and lightweight models, while enabling accuracy–efficiency trade-offs via an optional refinement module (AuxLoc). More broadly, explicitly structured latent geometry offers a compact, interpretable foundation for scalable perception and spatial inference under resource constraints.

## 6. Limitations and future work

WorldComp2D is validated mainly on a controlled 2D landmark localization setting, so generalization to more diverse objects, clutter, occlusion, and larger viewpoint/scale changes remain uncertain. The method also assumes a fixed observation policy (periodic fixation points). Future work includes extending to 3D embodied settings, learning an adaptive fixation policy, and evaluating on broader benchmarks and edge-deployment targets.

## Acknowledgments

This research was supported by Institute of Information & communications Technology Planning & Evaluation (IITP) grants funded by the Koreagovernment (MSIT) (RS-2023-00229689 and RS-2023-00253914).

## Impact Statement

This paper presents work whose goal is to advance the field of machine learning. There are many potential societal consequences of our work, none of which we feel must be specifically highlighted here.

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

# A. Training PdEnc, Loc, and AuxLoc

*Table 1.* Hyperparameters used.

|  | PdEnc | Loc | AuxLoc |
|---|---|---|---|
| Optimizer | Adam | Adam | Adam |
| # Epochs | 2000 (600 on AFLW) | 1000 (400 on AFLW) | 1000 (400 on AFLW) |
| Batch size | 128 | 50 | 50 |
| Initial lr | 1E-2 | 5E-4 | 5E-4 |
| Lr decay | 0.1 | 0.1 | 0.1 |
| Decay epoch | 1000 (300 on AFLW) | 500 (200 on AFLW) | 500 (200 on AFLW) |

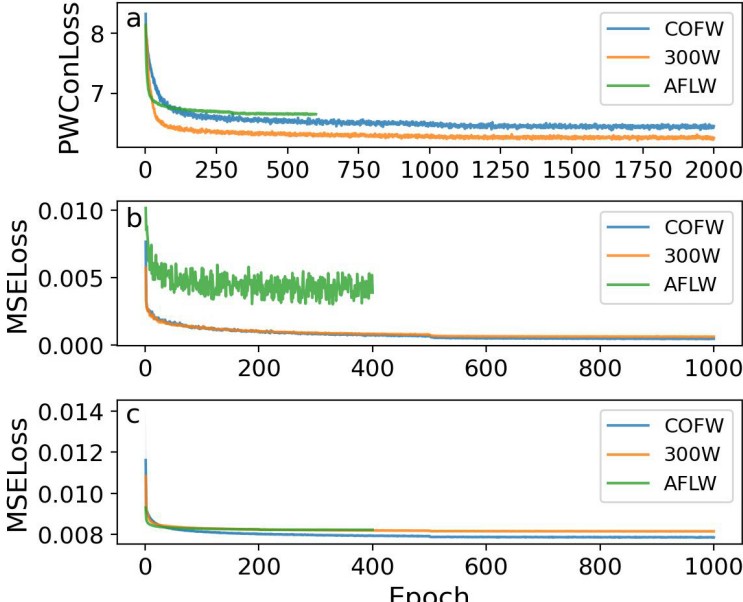

*Figure 1.* Learning curve for **(a)** PdEnc, **(b)** Loc, and **(c)** AuxLoc.

Table 1 lists the hyperparameters used to train PdEnc, Loc, and AuxLoc on the COFW, 300W, and AFLW datasets. The learning behavior of these elementary modules is plotted in Fig. 1.

# B. Summary of architectures for elementary modules

Table 2 lists the architectures of elementary modules on the COFW, 300W, and AFLW datasets.

*Table 2.* Network architectures.

| **PdEnc (C9)** |
|:---:|
| 2C9-9C16-16C32-32C64-FC256-FC128-L2Norm for COFW |
| 6C9-9C16-16C32-32C64-FC256-FC128-L2Norm for 300W and AFLW |
| **PdEnc (C16)** |
| 2C16-16C16($s = 1$)-16C32-32C32($s = 1$)-32C64-64C128-FC512-FC256-L2Norm for COFW |
| 6C16-16C16($s = 1$)-16C32-32C32($s = 1$)-32C64-64C128-FC512-FC256-L2Norm for 300W and AFLW |
| **PdEnc (C32)** |
| 2C32-32C32($s = 1$)-32C64-64C64($s = 1$)-64C128-128C256-FC512-FC256-L2Norm for COFW |
| 6C32-32C32($s = 1$)-32C64-64C64($s = 1$)-64C128-128C256-FC512-FC256-L2Norm for 300W and AFLW |
| **Loc** |
| FC512-FC512-FC$n$-Tanh ($n = 58$ for COFW, $n = 136$ for 300W, $n = 38$ for AFLW) |
| **AuxLoc** |
| 2C24-4*DW24-C1 for COFW |
| 4C24-4*DW24-C1 for 300W and AFLW |

Unless otherwise specified, all convolution layers in PdEnc have the same convolutional operational settings, $k_H = k_W = 3$, $s = 2$, $p = 1$.

