# OpenReview forum: "WorldComp2D: Spatio-semantic Representations of Object Identity and Location from Local Views"
_ICML.cc/2026/Conference — ICML 2026 regular_

### Official Review · Reviewer_RYAo · 2026-03-09

**Soundness:** 3
**Presentation:** 3
**Significance:** 2
**Originality:** 2
**Overall Recommendation:** 5
**Confidence:** 4

**Summary:**

This paper proposes WorldComp2D, a lightweight framework for learning fixation-centered spatio-semantic representations from local views, with the aim of jointly capturing object identity and relative spatial location under tight compute budgets. The key methodological idea is a proximity-weighted contrastive objective that preserves both class identity and 2D proximity structure in the latent space. The analysis is convincing on this representation-learning claim, showing clear class-wise clustering and latent distances that track real spatial distances. The downstream localization results are more mixed: the model is highly efficient and performs well on AFLW, but still trails stronger regression-based methods on COFW and 300W.

**Compliance With Llm Reviewing Policy:**

Affirmed.

**Final Justification:**

My concerns have been adequately addressed. In particular, the authors clarified the conceptual relation to prior object-centric “what/where” work and made clear that this connection will be reflected more explicitly in the paper’s positioning. The additional discussion around the lightweight two-patch fixation-centered design and the accuracy/efficiency trade-off also helps contextualize the method more convincingly. Overall, the rebuttal resolves my main concerns, and I therefore update my overall score from 3 to 5 (accept).

**Key Questions For Authors:**

1. How does the method relate conceptually to prior object-centric work that explicitly separates identity and location, and why is that literature not discussed in the current draft?
2. Did you test whether scaling up `PdEnc` / `Loc` reduces the COFW and 300W accuracy gap, or is the current ceiling mainly due to the local-view formulation rather than model capacity?
3. Would replacing the current two-patch setup (local patch plus context patch) with a fully foveated observation mechanism provide any benefit in representation quality or localization accuracy?

**Limitations:**

yes

**Strengths And Weaknesses:**

## Strengths
- The proximity-weighted contrastive loss is a compelling idea: it imposes spatial structure on the latent space while also encouraging class-level separation, which is more interesting than a standard binary contrastive setup.
- The representation analysis is one of the stronger parts of the paper. Experiments on intra-class clustering, inter-class separation, and latent-distance versus real-distance relationships make the claimed latent geometry concrete.
- The efficiency story is strong. With a much smaller model, the paper reports competitive AFLW accuracy, low parameter/FLOP counts, real-time CPU inference, and a clear accuracy-efficiency trade-off via the optional refinement stage.

## Weaknesses
- Related-work coverage should be strengthened. The paper makes a conceptual distinction between object identity and location, but does not discuss relevant object-centric work such as *Learning What and Where: Disentangling Location and Identity Tracking Without Supervision* [1]. Acknowledging that connection would help position the contribution more clearly.
- The paper should also engage more with local-observation / multiscale receptive-field methods. Recent foveated approaches that combine fine local detail with broader context, such as *Segment This Thing* [2] and *Looking Locally* [3], seem relevant to discuss even if the task setting differs.
- The quantitative localization results are mixed. The method is impressively small and outperforms some older baselines on AFLW, but it still underperforms stronger supervised regression methods on COFW and 300W. As a result, the broader performance claim feels narrower than the framing suggests, and it remains unclear whether a larger model would materially close the gap.
- Table 4 is useful, but it only shows how WorldComp2D degrades under corruptions. Without comparable degradation numbers for competing methods, it is difficult to assess whether the model is actually robust relative to baselines or simply internally stable.
- The evaluation scope is narrow relative to the embodied-AI framing. Because the method is validated mainly on facial landmark localization, it remains unclear how well the representation transfers to more general object-centric or embodied perception problems.

---
Overall, the paper presents a neat representation-learning idea and a strong efficiency profile, but the limited engagement with adjacent object-centric/foveated work and the mixed headline accuracy reduce the strength of the overall contribution.

---
[1] Manuel Traub, Sebastian Otte, Tobias Menge, Matthias Karlbauer, Jannik Thuemmel, and Martin V. Butz. *Learning What and Where: Disentangling Location and Identity Tracking Without Supervision*. arXiv:2205.13349, 2022. https://arxiv.org/abs/2205.13349

[2] Tanner Schmidt and Richard A. Newcombe. *Segment This Thing: Foveated Tokenization for Efficient Point-Prompted Segmentation*. arXiv:2506.11131, 2025. https://arxiv.org/abs/2506.11131

[3] Manuel Traub and Martin V. Butz. *Looking Locally: Object-Centric Vision Transformers as Foundation Models for Efficient Segmentation*. arXiv:2502.02763, 2025. https://arxiv.org/abs/2502.02763

---

> ### Author Rebuttal · Authors · 2026-03-31
>
> $\textbf{Q1. Relation of this work to prior object-centric work}$\
> $\textbf{A.}$ Conceptually, WorldComp2D shares the goal of prior object-centric work in isolating 'what' and 'where'. However, our method differs by anchoring this disentanglement to a fixation point, which is critical for agents with limited receptive fields. We did not discuss this specific literature in the current draft to prioritize work focused on the real-time, embodied AI domain, but we agree that acknowledging this connection clarifies our contribution in structuring latent geometry for local observations. If our paper is accepted, we promise to address this in the manuscript.
>
> $\textbf{Q2. Trade-off between model size and accuracy}$\
> $\textbf{A.}$ We can further increase the accuracy by enlarging the model size. During the rebuttal period, we have conducted addition experiments on COFW using larger PdEnc and Loc. The results are as follows.\
> | Model | NME on COFW | Number of params | FLOPs|\
> | PdEnc (C32) + Loc (baseline) | $5.16\pm0.05$ | 2.4M | 293.7M |\
> | PdEnc (C64) + Loc (baseline) | $5.01\pm0.02$ | 5.6M | 697.7M |\
> | PdEnc (C32) + Loc (1024) | $5.09\pm0.06$ | 4.3M | 297.2M |
>
> We have given up further increasing the model size to satisfy the real-time capability of our framework on CPU.
>
> $\textbf{Q3. Use of foveated observation}$\
> $\textbf{A.}$ Replacing the current setup with a fully foveated mechanism would likely offer benefits in representation smoothness and distance-resolution accuracy, especially for objects spanning the boundary of our current receptive fields.However, our two-patch approach ($o^{[1]}$ and $o^{[2]}$) was specifically designed as a discrete, lightweight approximation of foveation. By focusing on two scales, we maintain the strict computational efficiency required for real-time CPU operation on embodied agents, where patch extraction and resizing already constitute a significant portion of the processing time.We believe the current results demonstrate that this two-scale approach provides an optimal accuracy-speed trade-off, though we agree that investigating a more granular foveated hierarchy is a compelling direction for future work focused on higher-tier hardware targets.

---

> > ### Author Rebuttal · Reviewer_RYAo · 2026-04-01
> >
> > My concerns have been adequately addressed. In particular, the authors clarified the conceptual relation to prior object-centric “what/where” work and made clear that this connection will be reflected more explicitly in the paper’s positioning. The additional discussion around the lightweight two-patch fixation-centered design and the accuracy/efficiency trade-off also helps contextualize the method more convincingly. Overall, the rebuttal resolves my main concerns, and I therefore update my overall score from 3 to 5 (accept).

---

### Official Review · Reviewer_3wv2 · 2026-03-10

**Soundness:** 2
**Presentation:** 2
**Significance:** 2
**Originality:** 1
**Overall Recommendation:** 3
**Confidence:** 4

**Summary:**

WorldComp2D introduces a proximity-weighted contrastive encoder that processes a small set of multi-scale image patches centered around a fixation point. The encoder produces a spatio-semantic latent representation constrained to lie on a hypersphere. A learnable linear projection (defined by an anchor and direction) maps this representation to a line-manifold codebook , which is intended to capture both semantic identity and spatial proximity. A global localization module then predicts coordinates for all object classes from these latent vectors, and an optional refinement module  improves the predictions through a heat-map based adjustment.

**Compliance With Llm Reviewing Policy:**

Affirmed.

**Key Questions For Authors:**

How would the framework handle a dynamic or adaptive number of fixation points (for example, if an agent could actively choose where to look)?
Have you tested the method on non-facial keypoint tasks, such as human pose estimation or object-corner detection?
Could the proximity-weighted contrastive loss be combined with an explicit pose prior to improve performance on distance-related predictions?
What happens to both NME and computational cost when the codebook size is increased beyond 256 entries?
Are the learned anchor vectors pk stable across different random seeds, or do they vary substantially?

**Limitations:**

yes

**Strengths And Weaknesses:**

Strengths: Introduces a fixation-centered contrastive loss that explicitly incorporates spatial proximity in the weighting scheme. The experiments show that the learned latent space captures both intra-class clustering (Fig. 4a) and real-world distance relationships (Fig. 4c). The computational cost is very low (about 0.55M FLOPs for PdEnc and roughly 3M FLOPs for Loc), making real-time CPU inference feasible.
Weaknesses
1. The evaluation focuses only on facial-landmark localization, so it remains unclear how well the method generalizes to other perception tasks such as object detection, semantic segmentation, or 3-D pose estimation.
2. The approach assumes a fixed set of fixation points; agents operating in dynamic environments may require a more adaptive mechanism.
3. Compared with strong heat-map baselines (e.g., HRNet), the reported NME on COFW and 300-W is somewhat higher, indicating a trade-off in accuracy.
4. The paper does not compare with other lightweight localization approaches, such as depth-wise CNN architectures or transformer-based keypoint estimators.
5. The line-codebook representation might be limited when modeling highly non-linear visual patterns that cannot be well approximated by a first-order manifold.

---

> ### Author Rebuttal · Authors · 2026-03-31
>
> $\textbf{Q1. Dynamic and adaptive number of fixation points}$\
> $\textbf{A.}$ WorldComp2D is architecturally prepared for dynamic fixation. The Localizer's ability to aggregate information across variable $N_F$ values ($4, 5, 9$) demonstrates that it can already handle different levels of evidence. An adaptive agent would leverage the proximity-preserving nature of our latent space to choose fixation points that resolve spatial ambiguity, effectively moving from a fixed grid to an active search that prioritizes regions of high informational value.
>
> $\textbf{Q2. For other test vectors}$\
> $\textbf{A.}$ We chose facial landmark localization as a rigorous proof-of-concept because it provides a well-defined structure for evaluating spatio-semantic latent geometry. However, WorldComp2D is a general-purpose framework; its components are class-agnostic and designed to learn identity and proximity for any object set. We have tested our framework on the MSCOCO dataset and acquired the following preliminary results on  key point embeddings.
>
> $\textbf{Intra-class clustering}$\
> | Keypoint idx |   1    |   2    |   3    |   4    |   5    |   6    |   7    |   8    |   9    | \
> | L2-distance  | 0.172 | 0.159 | 0.160 | 0.171 | 0.172 | 0.231 | 0.231 | 0.257 | 0.258 | \
> | Keypoint idx |  10   |   11   |   12   |   13   |   14   |   15   |   16   |   17   |\
> | L2-distance  | 0.268 | 0.269 | 0.252 | 0.252 | 0.252 | 0.226 | 0.227 | 0.203 |
>
> $\textbf{Inter-class separation for a given key point (13)}$\
> | Keypoint idx |   1    |   2    |   3    |   4    |   5    |   6    |   7    |   8    |   9    | \
> | L2-distance  | 0.444 | 0.467 | 0.456 | 0.460 | 0.451 | 0.382 | 0.448 | 0.411 | 0.380 | \
> | Keypoint idx |  10   |   11   |   12   |   13   |   14   |   15   |   16   |   17   |\
> | L2-distance  | 0.489 | 0.354 | 0.377 | 0.000 | 0.503 | 0.368 | 0.505 | 0.494 |
>
> $\textbf{Spatial-distance (from key point 13)}$\
> | Spatial distance |   0   |   10   |   20   |   30   |   40   |   50   |   60   |   70   |   80   | \
> | L2-distance      | 0.252 | 0.268 | 0.289 | 0.312 | 0.333 | 0.356 | 0.385  | 0.421 | 0.458 |
>
> This preliminary results highlight successful key point embedding. Due to the lack of time, we have not evaluated the key point localization yet.
>
> $\textbf{Q3. Combining with explicit pose prior}$\
> $\textbf{A.}$ We agree that combining PWConLoss with an explicit pose prior is a compelling extension. While WorldComp2D currently learns these relationships implicitly through multiscale inputs and proximity-weighted loss, an explicit prior could act as a geometric regularizer. A regularization term could be added to the training objective that penalizes latent configurations that deviate significantly from a learned or predefined pose manifold. This would ensure that even when local visual evidence is weak, the latent representation $z$ remains constrained by a plausible geometric structure. Furthermore, the Localizer currently aggregates latent representations $(z, F)$ to infer coordinates. Providing an explicit pose prior as an additional input feature would allow the Localizer to use the learned spatio-semantic information to refine a baseline pose, rather than predicting coordinates from scratch.
> This would be particularly beneficial for distal objects where local visual evidence is insufficient for reliable proximity estimation. We view this as a potential enhancement for future work that maintains our commitment to lightweight, fixation-centered embodied perception.\
>
> $\textbf{Q4. What if codebook size exceeds 256?}$\
> $\textbf{A}.$ While the reviewer is correct that linear manifolds have limitations, WorldComp2D addresses this through a deep CNN encoder (PdEnc) that learns a highly non-linear mapping from pixel space to a hyperspherical latent manifold. By utilizing multiscale receptive fields, the model captures both high-frequency local details and low-frequency contextual cues, enabling it to resolve complex visual patterns such as large pose variations and partial occlusions. Furthermore, our proximity-weighted contrastive loss employs a non-linear exponential weighting function to ensure the latent space remains structured even when physical features undergo non-linear deformations.
>
> $\textbf{Q5. Effect of random seeds}$\
> $\textbf{A. }$ Yes, the emedding significantly depends on the random seed. However, it barely affects their relative geometric structure like intra-class clustering and inter-class separation and the resulting localization accuracy.

---

> > ### Author Rebuttal · Reviewer_3wv2 · 2026-04-05
> >
> > After carefully considering the authors’ rebuttal, I acknowledge that several points have been clarified. However, as some concerns remain only partially addressed, my overall assessment of the paper has not changed. Therefore, I will maintain my original score.

---

### Official Review · Reviewer_2FRy · 2026-03-11

**Soundness:** 4
**Presentation:** 3
**Significance:** 3
**Originality:** 3
**Overall Recommendation:** 6
**Confidence:** 1

**Summary:**

WorldComp2D aims to achieve efficient spatio-semantic inference from a small set of local observations. The framework consists of a proximity-dependent encoder, a localizer, and an optional auxiliary localizer. It introduces a proximity-weighted contrastive loss to address the limitation of traditional binary contrastive learning in capturing continuous affinity values, thereby enabling latent space distances to reflect fixation-centered spatial proximity. The method is evaluated on facial landmark localization tasks, demonstrating substantial computational efficiency gains at a modest cost in localization accuracy.

**Compliance With Llm Reviewing Policy:**

Affirmed.

**Final Justification:**

The authors' response addressed my questions in great detail. Overall, I believe this is a solid paper that can be accepted by the conference.

**Key Questions For Authors:**

1. Beyond facial landmark localization, I am interested in understanding the potential application scenarios and challenges of applying WorldComp2D to other embodied AI tasks, such as navigation or grasping.
2. For cases involving nested or overlapping objects, I would like to know how WorldComp2D handles such complex spatial relationships.

**Limitations:**

yes

**Strengths And Weaknesses:**

Strengths

1. The manuscript provides a clear description of the proposed network architecture.
2. It presents credible experimental results that validate the effectiveness of the approach.
3. The paper analyzes the limitations of traditional contrastive learning methods in handling continuous and fine-grained relationships between samples, and proposes a solution using proximity-weighted contrastive learning.

Weaknesses

1. The network architecture could be represented more intuitively using diagrams or tables, rather than long, cryptic string notations.
2. The abstract appears to be overly long.

---

> ### Author Rebuttal · Authors · 2026-03-31
>
> $\textbf{W1/2. Network presentation and abstract length}$\
> $\textbf{A}.$ We thank the reviewer for the suggestions. Unfortunately, we cannot revise the manuscript for the moment. If our paper is accepted, we promise to accordingly revise our manuscript.
>
> $\textbf{Q1. Potential application scenarios and challenges}$\
> $\textbf{A}.$ We propose the following application scenarios and consequent challenges\
> $\textbf{Potential application scenarios}$\
> $\bullet$ $\textbf{Landmark-based navigation}:$ In navigation, WorldComp2D can represent environmental anchors like doors, furniture, or some key objects in the spatio-semantic latent space. Because latent distances reflect real-world proximity, an agent can infer its relative distance to these landmarks to perform local self-localization or topological mapping.\
> $\bullet$ $\textbf{Part-based grasping}:$ For grasping, the framework can be applied to identify specific functional parts of an object. The multiscale receptive fields (RF1 and RF2) allow the agent to capture the fine-grained texture for precise contact points while maintaining the broader contextual information of the object's orientation.\
> $\textbf{Challenges}$\
> $\bullet$ $\textbf{3D Generalization}:$ The current framework is validated in 2D settings. Transitioning to grasping or navigation in the 3D real world would require the latent space to account for depth and 3D spatial proximity, which remains a key area for future work.\
> $\textbf{Dynamic fixation policy}:$ Currently, the model uses a fixed periodic fixation policy ($N_F=9$). For efficient navigation or grasping, the agent would likely need an adaptive policy to "choose" fixation points based on visual uncertainty or task relevance.
>
> $\textbf{Q2. For nested or overlapping objects}$\
> $\textbf{A}.$ If the key points for each of overlapping objects are well defined, we believe that our framework can successfully estimate several hidden key points by other objects with regard to its robustness to occluded facial landmarks shown in Table 4. However, for nested objects, it is desired to introduce a key point hierarchy and compare key points on the same level like face, left and right arms, and left and right legs to localize a face, but not left eye and face.

---

> > ### Author Rebuttal · Reviewer_2FRy · 2026-04-06
> >
> > Thanks for your detailed response. I have gained a deeper understanding of your manuscript and will keep my score.

---

### Official Review · Reviewer_svho · 2026-03-13

**Soundness:** 2
**Presentation:** 2
**Significance:** 1
**Originality:** 2
**Overall Recommendation:** 2
**Confidence:** 3

**Summary:**

This paper presents WorldComp2D, an efficient representation-learning framework tailored for embodied AI agents to perceive the environment in which they are deployed through local, viewpoint-dependent observations. The proposed framework explicitly structures the latent space to learn distances between representations that reflect real-world spatial proximity to a fixation point.

**Compliance With Llm Reviewing Policy:**

Affirmed.

**Key Questions For Authors:**

See Weakness.

**Limitations:**

The statement regarding negative societal impact is not presented.

**Strengths And Weaknesses:**

**Strengths**

- The idea of separating the spatio-semantic representation sounds interesting.

- The efficiency of the proposed method is remarkable, which reduces the FLOPS to MFLops Unit from GFlops Unit.

**Weaknesses**

Many aspects have already been mentioned in the last section; however, the reviewer would like to point out the weakness of the paper to justify the rating of this paper.

- Simplicity in the task: The scope of this paper is set to embodied agentic AI; however, the reviewer is concerned that the paper oversimplifies the problem in the name of exploring a new field, which may enable the light-weight architecture and improve efficiency. While the idea itself is interesting, the reviewer would like to see more coverage in the experiments.

- Discrepancy in the 2D tasks vs. embodied agents: The embodiment indicates deploying the agents in a 3D environment, whereas the proposed method is limited to the 2D tasks, specifically, face landmark localization.

- Fixation Dependency: The system relies on the strong assumption of a fixation point. This indicates the agent already recognizes the object and knows what to look at. In a truly robust perception system, the order is to first figure out where to fixate, and then operate on a specific case. If the fixation is poorly chosen (e.g, an image is 2D-rotated or occluded), the resulting representation would be less effective for the given task. Also, the chosen datasets assume the face is centered in the image.

- Dual-receptive field approach: The reviewer believes the concept of multiple receptive fields is already baked into modern deep learning architectures. Unlike the existing methods, the dual receptive field design is built upon the fixation point assumption, which makes the proposed method dependent on the fixation steps. The reviewer suggests conducting an experiment regarding the random fixation points.

- Argument on the separated identity and location: The reviewer found that a supporting experiment is required to back the separation of location and identity representations. Could there be any disentanglement analysis on the learned representation itself?

---

> ### Author Rebuttal · Authors · 2026-03-31
>
> $\textbf{W1. Simplicity in the task}$\
> $\textbf{A}$. The datasets (COFW, 300W, and AFLW) are widely used for facial landmark detection evaluation, and the SoTA algorithms depend on models with higher computational complexity than ours. Thus, we cannot agree on the point that the improvement that we achieved is due to the use of simple datasets.
>
> $\textbf{W2. Discrepancy in the 2D tasks vs. embodied agents}$\
> $\textbf{A}$. This is the limitation that we honestly address in the limitations and future work session.
>
> $\textbf{W3. Fixation Dependency}$\
> $\textbf{A}$. We are afraid of the reviewer’s misunderstanding. The agent does not recognize any objects in advance. We simply use a periodic fixation policy that chooses not what to look at but where to look at. Any objects are unnecessarily located on the fixation points. We have already identified the robustness of our method to occlusion cases as reported in Table 4.
>
> $\textbf{W4. Dual-receptive field approach}$\
> $\textbf{A}$. The reviewer is right. The number of iterative fixation steps is of great importance in latency and the number of FLOPs. The periodic fixation policy is simple but supports noniterative single batched feature extraction with reliable landmark localization. According to the Reviewer’s suggestion, we have conducted new experiments with random fixation points. The results on COFW are summarized in the table below, which highlights a large degradation of localization precision.
>
> |$\quad\quad\quad\quad\quad\quad\quad$| Periodic fixation policy (ous)  |  Random fixation points |\
> | NME on COFW$\quad$| $\quad\quad\quad\quad 5.16\pm 0.05\quad\quad\quad$ |$\quad\quad7.45\pm 0.17\quad\quad$ |
>
> $\textbf{W5. Argument on the separated identity and location}$\
> $\textbf{A}$. By placing the fixation point on each landmark, we can exclude the contribution of fixation locations to the spatio-semantic representation and acquire the representation by identity. The results are shown in Figure 4, where we evaluate the identity representation for each landmark by means of L2-distance from the representation for a given landmark. The contribution of location to the spatio-semantic representation is evaluated by measuring the L2-distance change upon the distance between a fixation point and a given landmark as also shown in Figure 4.

---

> > ### Author Rebuttal · Reviewer_svho · 2026-04-03
> >
> > The reviewer would like to appreciate the authors for the clarification. However, the rebuttal partially addresses my concerns.
> >
> > A1. This still cannot address the problem. First of all, the scale of the models overall tends to grow these days. Also, to counterargue the question, there needs to be results on different tasks that demonstrate the effectiveness of the proposed algorithm. The credit is given to the configuration that the model has not been trained on the target dataset. The reviewer is still concerned by the scope of the paper, which is a bit narrow, even if acknowledged in the limitations section.
> >
> > A2. The reviewer respects the honesty of the authors, mentioning that the task itself is not well aligned with the embodied AI. Although this is addressed in the limitation section, the statement in this paper (Embodied AI) is still not well aligned with the task itself (2D face landmark detection). It appears that this paper has framed its topic to align with the current growing interest in the field of Embodied AI. Under the authors' rationale, one could argue that conventional classification or other recognition tasks might also be broadly categorized as Embodied AI if mentioned in the limitations section of the paper.
> >
> > A3-1. Would the occlusion size=40 indicate the edge of the square patch occlusion? The reviewer requires further clarification on how the occlusion was generated. Would it be removing the patch values? Can the authors provide a comparison of the other models' robustness to the image degradation that the authors evaluated the proposed model for?
> >
> > A3-2. Even though the augmentation of $\pm 10^\circ$ could not guarantee the extreme case, where the human lies down on the floor, this point is followed up in the A4.
> >
> > A4. The authors argue that the agent does not recognize the object in advance, where the dataset has well-centered faces with an upright pose. This renders the proposed fixation policy less reasonable since the faces will not appear in upright poses every time in the realistic scenario.
> >
> > A5. The reviewer thanks the authors for the clarification.
> >
> > After further clarification in A3-1, I will modify my rating.

---

> > > ### Author Response · Authors · 2026-04-06
> > >
> > > $\textbf{A1/A2. Narrow scope of the present work and alignment with the embodied AI scope.}$\
> > > $\textbf{A.}$ We appreciate the Reviewer’s feedback regarding the scope and the Embodied AI framing. We chose a facial landmark detection task as a PoC task to highlight the high-precision localization capability of our lightweight framework in comparison to rich benchmark results. We apologize for not addressing the full scope of our framework. We consider the hierarchy of landmarks (keypoints) with reference to semantic coverage in which a set of the facial landmarks is a subset of a face. In this scenario, the agent first finds faces using our framework from its two-scale local views and places its fixation on the face center using the localizer. Our framework could successfully be applied to human body key point localization using the COCO dataset (with 17 key points including face key points). We report our preliminary results on embedding the 17 key points using the PdEnc of the same architecture as for the facial landmark encoding.\
> > > \
> > > $\textbf{Intra-class clustering}$\
> > > | Keypoint idx | 1 (nose) | 2 (left eye) | 3 (right eye) | 4 (left ear) | 5 (right ear) | 6 (left shoulder) | 7 (right shoulder) | 8 (left elbow) | 9 (right elbow) |\
> > > | L2-distance | 0.172 | 0.159 | 0.160 | 0.171 | 0.172 | 0.231 | 0.231 | 0.257 | 0.258 |\
> > > | Keypoint idx | 10 (left wrist) | 11 (right wrist) | 12 (left hip) | 13 (right hip) | 14 (left knee) | 15 (right knee) | 16 (left ankle) | 17 (right ankle) |\
> > > | L2-distance | 0.268 | 0.269 | 0.252 | 0.252 | 0.252 | 0.226 | 0.227 | 0.203 |
> > >
> > > $\textbf{Inter-class separation for a given key point (13: right hip)}$\
> > > | Keypoint idx | 1 | 2 | 3 | 4 | 5 | 6 | 7 | 8 | 9 |\
> > > | L2-distance | 0.444 | 0.467 | 0.456 | 0.460 | 0.451 | 0.382 | 0.448 | 0.411 | 0.380 |\
> > > | Keypoint idx | 10 | 11 | 12 | 13 | 14 | 15 | 16 | 17 |\
> > > | L2-distance | 0.489 | 0.354 | 0.377 | 0.000 | 0.503 | 0.368 | 0.505 | 0.494 |
> > >
> > > $\textbf{Spatial-distance (from key point 13 (right hip))}$\
> > > | Spatial distance | 0 | 10 | 20 | 30 | 40 | 50 | 60 | 70 | 80 |\
> > > | L2-distance | 0.252 | 0.268 | 0.289 | 0.312 | 0.333 | 0.356 | 0.385 | 0.421 | 0.458 |
> > >
> > > $\textbf{Embodied AI rationale}$: The keypoint (landmark) detection is based on the agent’s local views rather than global images. The agent can find faces only from its two-scale local view using the framework. This will not require many iterations since the larger-scale view’s spatial coverage is quite large. Once the face is localized, the subset facial landmarks are localized using another pair of PdEnc and Loc that are detailed in the present manuscript. While we acknowledge the Reviewer's concerns in A2, 3-2, and 4 regarding the full operational pipeline, we chose this focused scope to provide a rigorous, benchmarked comparison against existing high-precision models. The success of the framework on these fine-grained landmarks serves as a proof-of-concept for its broader application to general object-centric tasks. We strategically focused the present manuscript on facial landmark detection to highlight the high-precision capabilities of our framework. Facial landmarks represent a complex, structured set of proximal objects that require fine-grained spatio-semantic reasoning to resolve.
> > >
> > > $\textbf{A3-1. Details of occlusion and comparison with previous works.}$\
> > > $\textbf{A.}$ The Reviewer is correct; the occlusion (size=40) refers to a 40×40 pixel square patch filled with zeros. One 40x40 pixel square patch is placed at a random coordinate on each test sample to simulate removed information.
> > > While few papers report robustness to synthetic occlusion, we compared our results against a relevant study (Yan et al., Pattern Recognition 116 (2021):107945) using the 300W dataset. This work applies an ellipse-shaped occlusion patch to each test sample. The ellipse-shaped patch used is slightly smaller than a 30x30 patch, so that we compare the degradation due to occlusion as follows.\
> > >
> > > | Method | Dataset | Degradation | NME |\
> > > | Ours$\ \ $ | 300W | No occlusion | 5.06 $\pm$ 0.01 |\
> > > |$\qquad\ \ \ \ $|$\qquad\ \ $       | Occlusion (size=30)  5.36 $\pm$ 0.01 (-5.9%) |\
> > > | HRNet| 300W | No occlusion | 3.32 |\
> > > |$\qquad\ \ \ \ $|$\qquad\ \ \ $| Occlusion | 3.57 (-7.5%) |
> > >
> > > Even compared to high-complexity models like HRNet, WorldComp2D shows a smaller relative performance degradation, suggesting that our explicitly structured latent space is more resilient to missing local features.
> > >
> > > $\textbf{A3-2/A4. Face-awareness in advance.}$\
> > > $\textbf{A.}$ We acknowledge the Reviewer’s concern regarding this point. We believe this is thoroughly addressed in our combined response to A1 and A2, where we detail the framework’s generalization and its alignment with embodied AI principles.

---

### Decision · Program_Chairs · 2026-04-30

**Decision:**

Accept (regular)

**Comment:**

This paper introduces WorldComp2D, a highly efficient representation learning framework designed to capture both object identity and spatial proximity from local, fixation-centered observations. The core contribution is a proximity-weighted contrastive loss that explicitly structures the latent space geometry. The method is primarily evaluated on 2D facial landmark localization tasks, demonstrating strong efficiency (reductions in FLOPs and parameter counts) and real-time CPU performance. The reviewers reached a mixed scores even after the rebuttal phase (2, 6, 3, 5). The AC recommends weak accept.